# FP-DETR: Detection Transformer Advanced by Fully Pre-training

**Wen Wang**[1*]**, Yang Cao**[1,2†]**, Jing Zhang**[3]**, Dacheng Tao**[4,3]
[1]University of Science and Technology of China
[2]Institute of Artificial Intelligence, Hefei Comprehensive National Science Center
[3]The University of Sydney, [4]JD Explore Academy, China
`wangen@mail.ustc.edu.cn, forrest@ustc.edu.cn,`
`jing.zhang1@sydney.edu.au, dacheng.tao@gmail.com`

## Abstract

Large-scale pre-training has proven to be effective for visual representation learning on downstream tasks, especially for improving robustness and generalization. However, the recently developed detection transformers only employ pre-training on its backbone while leaving the key component, i.e., a 12-layer transformer, being trained from scratch, which prevents the model from above benefits. This separated training paradigm is mainly caused by the discrepancy between the upstream and downstream tasks. To mitigate the issue, we propose FP-DETR, a new method that **F**ully **P**re-Trains an encoder-only transformer and smoothly fine-tunes it for object detection via a task adapter. Inspired by the success of textual prompts in NLP, we treat query positional embeddings as visual prompts to help the model attend to the target area (prompting) and recognize the object. To this end, we propose the task adapter which leverages self-attention to model the contextual relation between object query embedding. Experiments on the challenging COCO dataset demonstrate that our FP-DETR achieves competitive performance. Moreover, it enjoys better robustness to common corruptions and generalization to small-size datasets than state-of-the-art detection transformers. Code will be made publicly available at https://github.com/encounter1997/FP-DETR.

## 1 Introduction

Since the surge of deep learning-based object detector (Girshick et al., 2014), pre-training the backbone on large-scale datasets like ImageNet (Deng et al., 2009) and fine-tuning the model on downstream tasks has become a well-established paradigm. Pre-training significantly improves robustness (Hendrycks et al., 2019) and may enhance the model performance, especially on small datasets.

Following this paradigm, modern object detectors, like Faster RCNN (Ren et al., 2016) and YOLO (Redmon et al., 2016), generally add a *lightweight* task-specific layers on top of ImageNet pre-trained backbones, so that the model can enjoy the aforementioned benefits of pre-training. However, the recently developed detection transformers[1], e.g., DETR (Carion et al., 2020) and Deformable DETR (Zhu et al., 2020), only pre-trains its CNN backbone, while leaving the core module, *a 12-layer transformer* (including both encoder layers and decoder layers), trained from scratch in downstream tasks. As a result, the model suffers from limited robustness against common corruptions and the reliance on a large amount of training data for fine-tuning. Though UP-DETR (Dai et al., 2020) attempts to mitigate this problem by unsupervised pre-training, it requires an off-the-shelf CNN backbone that has already be pre-trained. Moreover, the two-stage separate pre-training methods may impair the model performance, since the two parts work together for object detection during fine-tuning.

The key reason that hinders existing detection transformers from benefiting from large-scale pre-training can be attributed to the discrepancy between the upstream task and the downstream task.

---

[*]This work was done during Wen Wang's internship at JD Explore Academy. [†]Corresponding author.
[1]In this paper, we use detection transformer represents a class of object detectors built upon transformers, while DETR represents the seminal work by (Carion et al., 2020).

Firstly, the object detection-oriented model structures are difficult to adapt to the ImageNet classification task. For example, the transformer decoder requires multiple query embeddings for detecting objects, while for ImageNet classification, there is only a single query embedding (class token) used. If the decoder is included during pre-training, both the self-attention layers and the projections on query embeddings in cross-attention layers may easily overfit to the single class token, making it is hard to pre-train the decoder. Secondly, the upstream ImageNet classification task misses some crucial components for the downstream object detection task. For example, while the downstream object detection task requires both localization and classification for the objects of interest, the upstream classification task only focuses on the latter. Moreover, the object relation modeling in object detection, which is important for removing heuristic post-processing like non-maximum suppression (NMS) (Carion et al., 2020), is absent in image classification.

To mitigate these problems, we propose FP-DETR, a novel method that reformulates the pre-training and fine-tuning phases for detection transformers. It fully pre-trains a detection transformer on ImageNet classification task and smoothly fine-tune it for object detection task though a task adaptor. Concretely, during pre-training, we introduce an encoder-only transformer structure by removing the decoder that can hardly be well pre-trained on the ImageNet classification task. Moreover, since both the CNN backbone and the transformer encoder in detection transformers can be seen as feature extractor, we replace the complex CNN backbone with a simple multi-scale tokenizer and only uses the transformer encoder for feature extraction. The resultant architecture is an efficient transformer encoder-only model. During fine-tuning on the object detection task, we take inspiration from the success of textual prompt (Liu et al., 2021a) in NLP and treat the query positional embeddings as visual prompts to help the model attend to the target areas (prompting) and recognize the object. Our key intuition is that if the model knows where to look at, it simply needs to identify the category of the object in hand, as did during pre-training. Based on this motivation, we devise a lightweight task adaptor to enhance the prompt ability of query positional embeddings and hence bridge the gap between upstream classification task and downstream object detection task. It leverages self-attention to emphasize the relation between query embeddings, such that the inter-object relationships, which have been neglected during pre-training, can be captured during fine-tuning.

Experiments show that the proposed FP-DETR achieves competitive performance on the challenging COCO dataset (Lin et al., 2014), and a better trade-off between the number of parameters and detection accuracy. Moreover, FP-DETR is more robust against common corruptions and can generalize well to small-size datasets like Cityscapes (Cordts et al., 2016), attributing to the effective fully pre-training.

## 2  RELATED WORK

**Object Detection.** Object detection is one of the fundamental tasks in computer vision (CV). Representative object detection methods can be roughly categorized as the one-stage object detector, e.g., YOLO (Redmon et al., 2016), SSD (Liu et al., 2016), and two-stage object detectors like Faster-RCNN (Ren et al., 2016). While significant progress has been made, these methods are not end-to-end and rely on heuristic components like non-max suppression (NMS) and rule-based label assignments. Recently, DETR (Carion et al., 2020) formulates the object detection task as a set prediction problem and proposes an end-to-end pipeline to exploit transformer and bipartite matching for detection. The success of DETR brought the recent surge of detection transformers. For example, Deformable DETR (Zhu et al., 2020) proposes deformable attention to significantly accelerate the model convergence and reduce the computational cost, while allowing the model to benefit from multi-scale features. Conditional DETR (Meng et al., 2021) narrows down the spatial range for localizing the object regions via learning the decoder embedding conditioned on a spatial query. REGO-DETR (Chen et al., 2021) mitigate the training difficulty through Region-of-Interest (RoI) based detection refinement. While effective, the crucial component in these models, i.e., a 12-layer transformer, is trained from scratch, which limits the robustness and generalization ability of the detection models.

**Pre-training for Object Detection.** The seminal work RCNN (Girshick et al., 2014) demonstrates the many benefits of ImageNet pre-training for object detection, e.g., better detection performance and faster convergence. Since then, pre-training and fine-tuning has become a well-established paradigm for object detection. Recently, He et.al challenge the common belief by showing that

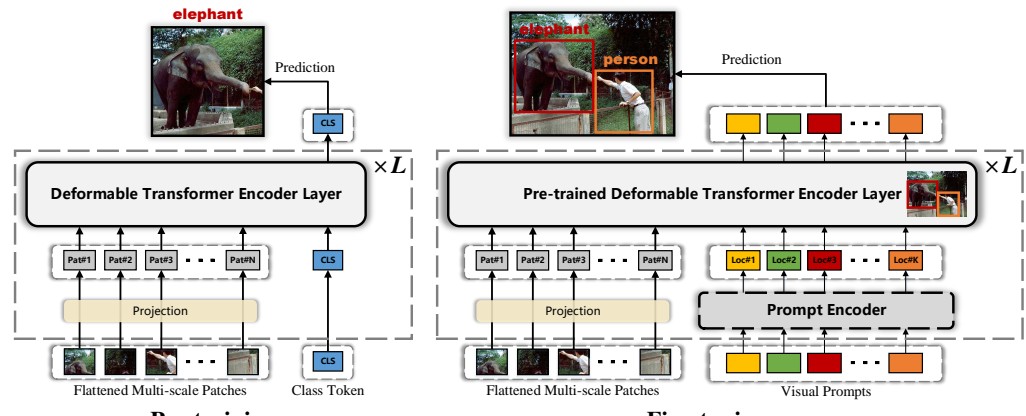

Figure 1: Illustration of pre-training and fine-tuning stages of FP-DETR.

training from scratch on datasets like COCO with a longer schedule can also produce a competitive detection performance (He et al., 2019). However, Hendrycks et.al show that though comparable in-domain performance can be achieved, training from scratch results in much worse out-of-domain generalization (Hendrycks et al., 2019). Inspired by the success of pre-training for CNN-based object detectors, UP-DETR (Dai et al., 2020) proposes a novel random query patch detection pre-training task to improve DETR's performance and convergence. Though progress has been made, their method relies on a pre-trained off-the-shelf CNN backbone. Moreover, the improvement is limited, since the CNN backbone and the transformer are pre-trained separately while they need to work jointly in the downstream task. Probably the most similar work to ours is YOLOS (Fang et al., 2021), which also pre-trains and fine-tunes an encoder-only transformer for object detection. However, it aims at demonstrating the feasibility of detecting objects using BERT (Devlin et al., 2018), with minimum inductive bias. As a result, the discrepancy between the upstream and downstream tasks, which significantly degenerates the model performance, is not considered in YOLOS. By contrast, we aim at maximizing the benefits of pre-training for the downstream object detection task. To this end, we treat query positional embeddings as visual prompts and devise a novel task adaptor to smoothly fine-tune the pre-trained model on downstream tasks. Moreover, since object detection task often requires high-resolution images as input, inductive bias like sparsity is introduced to reduce the computational costs.

**Prompt-based Learning in NLP.** Large-scale pre-trained models like GPT-3 (Brown et al., 2020) advance the field of NLP. However, such large-scale models are not designed for fine-tuning. To better utilize these models for downstream tasks, the "pre-train, prompt, and predict" paradigm emerges in the NLP community, and has drawn increasing attention (Liu et al., 2021a). The new paradigm reformulates the downstream task to mimic the task during pre-training with the help of a textual prompt, to bridge the gap between upstream and downstream tasks. Specifically, pioneer work introduces manually designed prompts to improve the fine-tuning (Schick & Schütze, 2020), or searches the textual prompt for downstream tasks (Gao et al., 2020; Jiang et al., 2020). Though improvements have been made, these prompts correspond to natural language phrases in discrete space can be suboptimal, since the neural networks are inherently continuous. To tackle this problem, P-tuning (Liu et al., 2021b) proposes to optimize contiguous prompt embedding to better close the gap between pre-training and fine-tuning. In this paper, instead of reformulating the downstream task, we propose a new perspective and treat the query positional embeddings in detection transformers as visual prompts. To this end, a task adaptor is devised to facilitate fine-tuning on the downstream task.

## 3 METHOD

In this section, we first revisit the preliminaries on detection transformers, then introduce the pre-training and fine-tuning phases of our method as shown in Figure 1. Through exploiting the large-scale pre-training on ImageNet, our detection transformer can enjoy better detection performance, improved robustness, and enhanced generalization.

## 3.1 PRELIMINARIES ON DETECTION TRANSFORMERS

Detection transformers generally consist of the following parts: a backbone, a transformer encoder, a transformer decoder, and a task-specific feed-forward network (FFN). The backbone $F$ can be decomposed as multiple stages, i.e., $F = f^l \circ \cdots \circ f^2 \circ f^1$, where each stage $f_i$ takes the output from the previous stage as input and outputs the down-sampled feature map at stage $i$. For an input image $I \in \mathbb{R}^{H \times W \times 3}$, the backbone extracts multi-scale features, which are flattened and projected to 1-D sequence representation. Afterwards, the sequence representation is added with position and level embedding to obtained $x \in \mathbb{R}^{N \times D}$, where $N = \sum_{l=1}^{L} HW/S_l^2$ is the sequence length, $S_l$ is the down-sampling rate at $l$-th feature level, and $D$ is the dimension of the feature embedding. The transformer general consists of both encoder and decoder, and takes the sequence representation extracted from the image and the query embeddings as input for context modeling. The final prediction is made by applying FFN to the object representations.

While Deformable DETR (Zhu et al., 2020) contains four different feature levels with different down-sampling rates, most other detection transformers, including DETR (Carion et al., 2020), Conditional DETR (Meng et al., 2021), contains only a single feature level.

## 3.2 PRE-TRAINING

**Encoder-only Transformer.** As described in Section 1, for image classification, only a single class token is required for context aggregation. The six decoder layers trained on only one class token is prone to overfitting. To tackle this problem, we remove the decoder during pre-training on the image classification task, which leads to an encoder-only transformer structure. We follow Deformable DETR (Zhu et al., 2020) to adopt multi-scale deformable attention with four different feature levels to build our model. It largely alleviates the slow convergence of DETR (Carion et al., 2020), with the help of inductive bias like sparsity.

To perform image classification, an additional class token added with corresponding positional embedding $x_{cls} \in \mathbb{R}^D$ is concatenated with the sequence feature, for aggregating global context from the image tokens. Thus the input to the encoder-only transformer $z_0$ can be written as,

$$z_0 = \left[ x_{cls}; x^1; x^2; \cdots ; x^N \right]. \tag{1}$$

Afterward, the input sequence feature is processed by $T$ different transformer encoder layers, each consists of a multi-scale deformable self-attention (MSDSA) layer and an MLP layer. Residual connections and layer norm (LN) are applied after each sub-layer, i.e.,

$$z'_t = \text{LN}(\text{MSDSA}(z_{t-1}) + z_{t-1}) \qquad t = 1 \ldots T, \tag{2}$$
$$z_t = \text{LN}(\text{MLP}(z'_t) + z'_t) \qquad t = 1 \ldots T. \tag{3}$$

Final predictions are made on class token at the $T$-th layer, i.e., $z_T^0$. During implementation, the class token is treated as an additional feature level, with only one resolution. Since multi-scale deformable attention assumes different levels of feature maps are spatially aligned, which does not hold for the class token in $x_{cls}$, attention mask is utilized to remove the information flow from $x_{cls}$.

**Lightweight Multi-scale Tokenizer.** The transformer encoder's ability to serve as a general feature extractor has been proved by recent advances in vision transformers (Dosovitskiy et al., 2020). While existing detection transformers contain both sophisticated backbone and transformer encoder for feature extraction, we argue that such design brings unnecessary complexity to the detection model. Instead, we replace the complex backbone with simple convolutional layers that work as a multi-scale tokenizer. Specifically, the feature extractor at each stage $f_i$ is a simple single-layer convolutional layer that is designed only for down-sampling the feature map to expected resolution. Such design significantly simplifies the model and makes it easy to pre-train the detection model on ImageNet classification task. The detailed structure of our multi-scale tokenizer is shown in Appendix A.1.

**Discussion.** Most of the existing vision transformers adopt an encoder-only structure (Dosovitskiy et al., 2020; Liu et al., 2021c; Xu et al., 2021). By reformulating the detection transformer to an encoder-only structure, we solve the problem that the decoder is difficult to pre-train and simplify the model structure. Moreover, it allows us to take advantage of the existing experience in training vision transformers. In Appendix A.2, we explore pre-training on ImageNet using the encoder-decoder structure, however, the performance is much worse than that of the encoder-only structure.

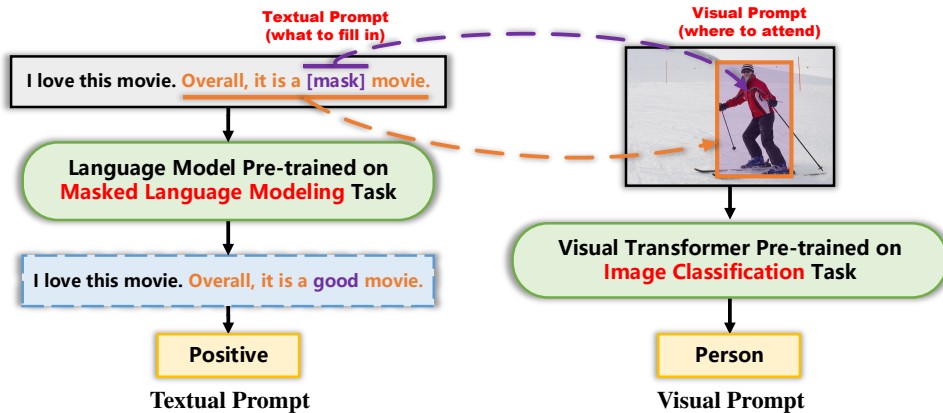

Figure 2: Analogy between textual prompt in NLP and visual prompt in CV.

### 3.3 FINE-TUNING

During fine-tuning, the single class token is replaced by $N_q$ query content embeddings (Carion et al., 2020) $x_{obj} \in \mathbb{R}^{N_q \times D}$ for object detection. The content embeddings are also added with query positional embeddings (Carion et al., 2020) $p \in \mathbb{R}^{N_q \times D}$ before feed into the transformer encoder. Thus the input to the transformer encoder is:

$$\mathbf{z}_0 = \left[ x_{obj}^1 + p^1; x_{obj}^2 + p^2; \cdots; x_{obj}^{N_q} + p^{N_q}; x^1; x^2; \cdots; x^N \right]. \tag{4}$$

The class token for image classification and the query content embeddings for object detection are both designed for context aggregation. While the former focuses on global context aggregation, the latter focus on aggregating the local context for one specific object instance.

**Query Positional Embeddings as Visual Prompts.** The textual prompt in NLP is shown in Figure 2. It reformulates the downstream task to mimic the pre-training one, so that the pre-trained model can better handle the downstream task. However, finding the optimal textual prompt is not easy, recent work such as P-tuning (Liu et al., 2021b) searches prompts in a contiguous space to bridge the gap between the upstream and downstream tasks.

Object detection can be decomposed into two subtasks, i.e., localization and classification. Our intuition is that if the classifier pre-trained on ImageNet knows where to look at, it can easily recognize the object within the specified region, as did during ImageNet classification pre-training. In our detection transformer, the query position embeddings are mapped to the reference points, which guide different query content embeddings to sample the corresponding context from the image content. In other words, the query position embedding works as a visual cue to point out the image region that the model should focus on, as shown in figure 2 (b). From this perspective, the query position embedding serves as a visual prompt, which is analogous to the textual prompt in NLP. Further, the process of training the model to localize objects corresponds to the process of searching textual prompts in contiguous space. And the process of final classification corresponds to the process of filling in the blanks and answers mapping in NLP (Liu et al., 2021a).

**Improving Fine-tuning with Task Adaptor.** Based on the above motivation, we propose a task adaptor to improve the query positional embeddings' ability to prompt. Specifically, the inter-object relationship can help the model better identify object regions. However, this crucial part is missing during ImageNet classification pre-training, since only one class token is used. To this end, we devise the task adapter to improve the modeling of the inter-object relationships. It pre-processes the visual prompts before they are sent to each pre-trained encoder layer, i.e.,

$$z_t^{0:N_q} = \text{TaskAdaptor}\left( z_t^{0:N_q} \right) \qquad\qquad t = 1 \ldots T. \tag{5}$$

By default, the task adaptor is instantiated as a self-attention layer. Afterward, the sequence is processed by the transformer encoder layers, as did in Equation 2 and 3. The object pattern embedding in the last encoder layer, i.e., $z_t^{0:N_q}$ is used for final prediction. Our task adaptor helps the visual

| Model | Layers | Hidden size $D$ | MLP size | Heads | Params | Acc@Top-1 |
|-------|--------|-----------------|----------|-------|--------|-----------|
| Lite | 12 | 256 | 1024 | 8 | 11M | 77.3 |
| Small | 12 | 384 | 1536 | 8 | 23M | 80.8 |
| Base | 12 | 480 | 1920 | 10 | 35M | 82.0 |

Table 1: Details of our FP-DETR variants. Parameters (Params) and top-1 accuracy (Acc@Top-1) here are calculated on the upstream ImageNet-1k (Deng et al., 2009) classification task.

prompts for individual objects to make associations with other instances, making them more suitable for the downstream object detection task.

**Discussion.** Recently, Shen et al. (2021) find the connection between named entity recognition (NER) in NLP and object detection in CV, and propose a two-stage method for NER. Similarly, We propose a new perspective to view query positional embeddings as visual prompts. However, the existing pre-trained visual models are not as powerful as pre-trained language models (Brown et al., 2020). As a result, we are unable to fix the pre-trained models and only tune the visual prompts, to perform few-shot or even zero-shot learning on downstream tasks, as did in NLP. But still, this new perspective helps us to better understand the detection transformers and the discrepancy between the pre-training and downstream tasks, which further helps us design the task adaptor to smoothly fine-tune the pre-trained model on downstream tasks.

## 4 EXPERIMENTS

### 4.1 IMPLEMENTATION DETAILS

**Datasets.** Following the common practice, our detector is pre-trained on ImageNet (Deng et al., 2009) and fine-tuned on COCO 2017 (Lin et al., 2014) train set. Evaluation results on the val set of COCO 2017 are reported. To evaluate the models' robustness against common corruptions, we report the model performances on COCO-C (Michaelis et al., 2019), which is obtained by applying corruption synthesis algorithm to COCO. Besides, we evaluated the model's generalization ability by fine-tuning on the small-size dataset, i.e., Cityscapes dataset (Cordts et al., 2016).

**Model Variants** The lite version of our model follows the hyper-parameters of the Deformable Transformer in Deformable DETR (Zhu et al., 2020). Specifically, the encoder-only transformer contains 12 self-attention layers, each layer with 8 heads, the dimension of each head is 32, and the dimension of the MLP layer in FFN is 1024. Furthermore, we introduce two other model variants by changing the number of heads, or dimensions per head, while keeping the model depth unchanged. Table 1 summarizes the different variants of our model.

**Training & Fine-tuning.** By default, our FP-DETR is pre-trained on ImageNet-1k (Deng et al., 2009) for 300 epochs with AdamW (Loshchilov & Hutter, 2018) optimizer and cosine learning rate scheduler. Training strategies in DeiT (Touvron et al., 2021a) are adopted, and the image size is set as $224 \times 224$. We use a batch size of 1,024 for training, and the initial learning rate is set as $5 \times 10^{-4}$. After pre-training, models are fine-tuned for 50 epochs with AdamW optimizer on the downstream tasks. The learning rate is initialized as $1 \times 10^{-4}$ and decreased by a factor of 0.1 at the 40th epoch. We follow the implementation of Deformable DETR to apply deep supervision (Lee et al., 2015) on the last six encoder layers. The class token and the query content embedding are concatenated with the image tokens at an intermediate (the 7th) encoder layer, following CaiT (Touvron et al., 2021b). Besides, we set both the number of sampling points and the feature levels in multi-scale deformable attention as 4, and the number of object query embeddings as 300. Models are fine-tuned with a batch size of 32. All experiments are implemented on the NVIDIA A100 GPU.

### 4.2 COMPARISON WITH STATE-OF-THE-ARTS

As can be seen in Table 2, FP-DETR achieves competitive performance on COCO 2017 val set. Specifically, FP-DETR-Base is comparable with state-of-the-art Deformable DETR, while its small variant with 24M parameters can match the performance of UP-DETR and outperforms Conditional DETR and DETR with about 40M parameters. It performs better on detecting small objects and has better localization ability.

Table 2: Comparision of FP-DETR with other detection transformers on COCO 2017 val set. Models are categorized as encoder-only (Enc) and encoder-decoder (Enc-Dec) according to the transformer structure. † indicates the model is pre-trained on ImageNet-21k.

| Method | Structure | Backbone | Epochs | AP | $AP_{50}$ | $AP_{75}$ | $AP_S$ | $AP_M$ | $AP_L$ | Params |
|---|---|---|---|---|---|---|---|---|---|---|
| DETR | Enc-Dec | ResNet-50 | 500 | 42.0 | 62.4 | 44.2 | 20.5 | 45.8 | 61.1 | 41M |
| UP-DETR | Enc-Dec | ResNet-50 | 300 | 42.8 | 63.0 | 45.3 | 20.8 | 47.1 | 61.7 | 41M |
| Conditional DETR | Enc-Dec | ResNet-50 | 50 | 40.9 | 61.8 | 43.3 | 20.8 | 44.6 | 59.2 | 44M |
| Deformable DETR | Enc-Dec | ResNet-18 | 50 | 40.1 | 58.4 | 43.7 | 22.0 | 43.4 | 53.0 | 23M |
| Deformable DETR | Enc-Dec | ResNet-34 | 50 | 42.3 | 60.7 | 46.0 | 24.2 | 45.8 | 56.1 | 33M |
| Deformable DETR | Enc-Dec | ResNet-50 | 50 | 43.8 | 62.6 | 47.7 | 26.4 | 47.1 | 58.0 | 40M |
| YOLOS-S | Enc | - | 150 | 36.1 | 56.5 | 37.1 | 15.3 | 38.5 | 56.2 | 31M |
| YOLOS-B | Enc | - | 150 | 42.0 | 62.2 | 44.5 | 19.5 | 45.3 | 62.1 | 127M |
| FP-DETR-Lite | Enc | - | 50 | 37.2 | 56.5 | 40.4 | 21.7 | 40.0 | 48.6 | 11M |
| FP-DETR-Small | Enc | - | 50 | 42.5 | 62.6 | 45.9 | 25.3 | 45.5 | 56.9 | 24M |
| FP-DETR-Base | Enc | - | 50 | 43.3 | 63.9 | 47.7 | 27.5 | 46.1 | 57.0 | 36M |
| FP-DETR-Base† | Enc | - | 50 | 43.7 | 64.1 | 47.8 | 26.5 | 46.7 | 58.2 | 36M |

Moreover, FP-DETR shows a better trade-off between model parameters and detection accuracy compared to Deformable DETR. For example, FP-DETR-Small outperforms Deformable DETR with ResNet-18 (He et al., 2016) backbone, which has a similar amount of parameters. It also slightly outperforms Deformable DETR with ResNet-34 backbone, which has about 10M more parameters. The smallest version of our model, FP-DETR-lite, achieves 37.2 mAP on COCO 2017 with only 11M parameters.

Besides, FP-DETR significantly outperforms YOLOS-Base which also adopts a transformer encoder-only structure, in terms of both model efficiency and detection accuracy. This is mainly due to the inductive bias introduced in our model as Deformable DETR and the prompt-inspired task adaptor for more effective fine-tuning, which better bridges the gap between pre-training and fine-tuning.

The last row shows the result of our FP-DETR-Base fine-tuned from ImageNet-21 pre-training, denoted as FP-DETR-Base†. As can be seen, FP-DETR can benefit from pre-training at a larger scale, and FP-DETR-Base† with 36M parameters matches the performance of Deformable DETR with 40M parameters. In Appendix A.3, we also show the result of Deformable DETR with ResNet-50 backbone pre-trained on ImageNet-21k. Interestingly, the result is slightly worse than Deformable DETR with ImageNet-1k pre-trained backbone. We conjuncture that only pre-training part of the model makes it hard for the randomly initialized transformer adapts to the pre-trained CNN backbone, especially when the backbone has already been well-trained.

Table 3: Ablations on the task adaptor on COCO 2017 val set.

| Task Adaptor | Shared | AP | $AP_{50}$ | $AP_{75}$ | $AP_S$ | $AP_M$ | $AP_L$ | Params |
|---|---|---|---|---|---|---|---|---|
| w/o | N/A | 32.8 | 50.9 | 35.2 | 18.2 | 35.8 | 43.6 | 11M |
| Bi-LSTM | | 34.6 | 52.9 | 37.3 | 19.1 | 38.1 | 45.4 | 11M |
| Bi-LSTM | ✓ | 34.9 | 53.0 | 37.5 | 20.2 | 38.0 | 46.9 | 11M |
| Self-attention | | 36.8 | 56.0 | 39.5 | 20.9 | 39.8 | 49.1 | 12M |
| Self-attention | ✓ | 37.2 | 56.5 | 40.4 | 21.7 | 40.0 | 48.6 | 11M |

## 4.3 ABLATIONS ON THE TASK ADAPTOR

To gain a better understanding of the effectiveness of the proposed task adaptor, we perform ablation studies on the lite version of FP-DETR, as shown in Table 3. We have the following observations. First, removing the task adaptor results in significant drops in model performance. Without the task adaptor, the model can not well capture the inter-object relationships, which is crucial for removing duplicates and improving object recognition. Second, the task adaptor instantiated with a bidirectional-LSTM layer also helps to adapt the pre-trained model to the downstream task, but it works slightly worse than a self-attention layer. We conjuncture that the bidirectional-LSTM is

slightly worse on modeling long-range dependency compared to the self-attention layer, especially when the number of query embeddings reaches 300. By default, we adopt the self-attention layer for the task adaptor. In this way, our FP-DETR is a pure transformer encoder structure. Third, interestingly, a single task adaptor shared by visual prompts from different transformer encoder layers performs slightly better than specialized task adaptors for each layer. Since the task adaptor is trained from scratch on the downstream task, we conjuncture that the shared task adaptor sees diverse data from different levels compared to the non-shared ones, and is thus more sufficiently trained. These experiments validate the effectiveness of our task adaptor on modeling object relationships, which is crucial for object detection.

Table 4: Comparison of model robustness on the COCO-C dataset.

| Method | Mean | Noise | | | Blur | | | |
|---|---|---|---|---|---|---|---|---|
| | | Gauss | Shot | Impul | Defocus | Glass | Motion | Zoom |
| DETR | 19.1 | 16.8 | 16.5 | 12.7 | 20.0 | 13.8 | 17.7 | 6.5 |
| UP-DETR | 21.6 | 22.1 | 22.2 | 18.7 | 20.1 | 13.4 | 18.3 | 6.9 |
| Conditional DETR | 20.3 | 18.6 | 18.6 | 16.0 | 20.5 | 14.8 | 18.4 | 7.1 |
| Deformable-DETR | 20.7 | 19.8 | 19.6 | 16.4 | 20.1 | 13.8 | 17.9 | 6.6 |
| YOLOS-S | 18.9 | 15.3 | 15.1 | 14.0 | 19.4 | 15.7 | 20.2 | 7.1 |
| FP-DETR-lite | 18.9 | 17.0 | 17.0 | 15.2 | 18.6 | 15.1 | 17.5 | 6.5 |
| FP-DETR-Small | 22.8 | 20.3 | 20.3 | 18.1 | 22.3 | 18.3 | 21.2 | 8.0 |
| FP-DETR-Base | 23.7 | 22.5 | 22.7 | 20.4 | 22.9 | 18.9 | 21.7 | 8.0 |
| Method | | Weather | | | Digital | | | |
| | | Snow | Frost | Fog | Bright | Contrast | Elastic | Pixel | JPEG |
| DETR | | 16.4 | 21.4 | 29.0 | 35.0 | 18.8 | 23.2 | 18.3 | 19.8 |
| UP-DETR | | 19.3 | 24.2 | 31.6 | 36.9 | 21.4 | 23.0 | 20.7 | 24.6 |
| Conditional DETR | | 18.2 | 22.9 | 30.0 | 34.7 | 20.6 | 24.0 | 19.4 | 20.5 |
| Deformable-DETR | | 18.3 | 23.6 | 31.7 | 37.0 | 21.2 | 24.0 | 19.3 | 21.0 |
| YOLOS-S | | 17.7 | 21.2 | 26.5 | 29.5 | 19.7 | 22.8 | 17.1 | 21.8 |
| FP-DETR-Lite | | 18.2 | 22.7 | 28.4 | 32.5 | 20.5 | 21.3 | 14.8 | 18.2 |
| FP-DETR-Small | | 22.9 | 27.1 | 33.1 | 37.3 | 25.0 | 25.5 | 20.2 | 23.1 |
| FP-DETR-Base | | 22.4 | 27.1 | 33.4 | 38.0 | 25.3 | 26.1 | 20.0 | 25.6 |

Table 5: Comparison of model generalization on the small-size Cityscapes dataset.

| Method | Structure | Epochs | AP | $AP_{50}$ | $AP_{75}$ | $AP_S$ | $AP_M$ | $AP_L$ | Params |
|---|---|---|---|---|---|---|---|---|---|
| DETR | Enc-Dec | 500 | $15.9_{\pm0.9}$ | $34.8_{\pm1.6}$ | $12.7_{\pm1.0}$ | $2.9_{\pm0.1}$ | $13.5_{\pm0.7}$ | $33.8_{\pm2.1}$ | 41M |
| UP-DETR | Enc-Dec | 300 | $23.8_{\pm1.3}$ | $45.7_{\pm2.2}$ | $20.8_{\pm1.2}$ | $4.0_{\pm0.5}$ | $20.3_{\pm1.9}$ | $46.6_{\pm1.9}$ | 41M |
| Conditional DETR | Enc-Dec | 150 | $21.1_{\pm0.7}$ | $42.7_{\pm1.3}$ | $18.8_{\pm1.6}$ | $3.6_{\pm0.3}$ | $19.8_{\pm0.3}$ | $41.1_{\pm1.0}$ | 44M |
| Deformable DETR | Enc-Dec | 50 | $27.3_{\pm0.6}$ | $49.2_{\pm0.6}$ | $26.3_{\pm0.8}$ | $8.7_{\pm0.2}$ | $28.2_{\pm0.8}$ | $45.7_{\pm0.7}$ | 40M |
| YOLOS-S | Enc | 150 | $9.8_{\pm0.1}$ | $25.3_{\pm0.4}$ | $6.1_{\pm0.4}$ | $1.9_{\pm0.2}$ | $8.1_{\pm0.4}$ | $20.7_{\pm0.4}$ | 31M |
| FP-DETR-Lite | Enc | 50 | $26.7_{\pm0.6}$ | $49.1_{\pm0.8}$ | $25.7_{\pm0.7}$ | $9.7_{\pm0.7}$ | $28.6_{\pm0.6}$ | $42.8_{\pm0.7}$ | 11M |
| FP-DETR-Small | Enc | 50 | $28.6_{\pm0.2}$ | $52.0_{\pm0.6}$ | $26.9_{\pm0.8}$ | $10.1_{\pm0.6}$ | $30.2_{\pm0.6}$ | $46.3_{\pm0.7}$ | 24M |
| FP-DETR-Base | Enc | 50 | $29.6_{\pm0.5}$ | $53.6_{\pm0.9}$ | $28.4_{\pm0.4}$ | $11.2_{\pm0.7}$ | $30.9_{\pm0.8}$ | $47.4_{\pm1.2}$ | 36M |

## 4.4 ROBUSTNESS TO COMMON CORRUPTIONS

Model robustness (He & Tao, 2020) is critical to trustworthy AI applications like autonomous driving. To this end, we evaluate the object detectors' robustness against common corruptions on COCO-C (Michaelis et al., 2019). As shown in Table 4, all detectors suffer significant performance drops under this rigorous condition. However, FP-DETR suffers the least performance drop compared to existing detection transformers. Notably, FP-DETR-Base performs best on 14 out of 15 types of corruptions, though it has comparable performance to Deformable DETR on the clean COCO 2017 val set. This is because the pre-training on ImageNet helps FP-DETR learn more generalizable representation. This is coherent with the observation in Hendrycks et al. (2019).

The first two rows in Figure 3 provide some qualitative results of our FP-DETR-Base on the COCO-C dataset. As can be seen, the images are significantly degenerated due to the existence of various corruptions, like elastic transform, zoom blur, fog, etc. However, FP-DETR-Base still manages to produce plausible results under low visibility and large distortion, which manifests its robustness.

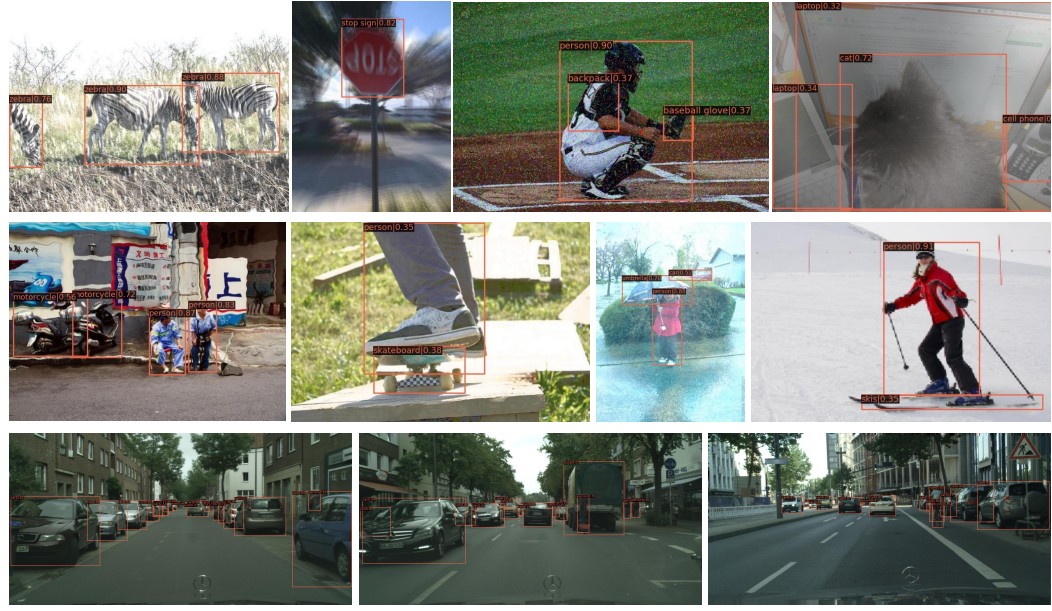

Figure 3: Qualitative detection results of FP-DETR on the COCO-C and Cityscapes datasets are shown in the first two rows and the last row, respectively.

## 4.5 GENERALIZATION TO SMALL-SIZE DATASET

In real-world applications, collecting a large amount of data is often infeasible. As a result, the models are required to perform well by training on only limited data. Table 5 shows the result of models fine-tuned on the Cityscapes dataset with only 2,975 training images. All models are trained with a batch size of 8 to guarantee enough training iterations. The results are averaged over five repeated runs with different random seeds. As can be seen, most detection transformers, including DETR, UP-DETR, Conditional DETR, and YOLOS-B, perform poorly under this condition. This is partly due to the lack of inductive bias in those models. As a result, the model requires a large amount of data to learn the sparsity that is beneficial for object detection on images. Deformable DETR performs better, thanks to the sparsity inductive bias introduced in the deformable trans-former. However, it still performs worse than our FP-DETR, since the transformer in Deformable DETR is trained from scratch. Specifically, both FP-DETR-Base and FP-DETR-Small outperform Deformable DETR with more parameters. Notably, our FP-DETR-Lite with only 11M parameters matches the performance of 40M Deformable DETR, attributing to the end-to-end fully pre-training of the transformer.

The last row in Figure 3 shows some qualitative results of our FP-DETR-Base on the Cityscapes dataset. As can be seen, FP-DETR quickly adapts to the small-size dataset, and produces accurate object detection results, even for the small cars in the distant area. These results demonstrate the generalization ability of our method.

## 5 CONCLUSION

In this paper, we propose a novel detection transformer named FP-DETR that can take advantage of pre-training on upstream datasets to enable better robustness and generalization. FP-DETR contains a simple encoder-only structure for fully pre-training and can be smoothly fine-tuned for object detection via a task adapter. It leverages query positional embeddings as visual prompts to help the model attend to the target area (prompting) and recognize the object, effectively mitigating the gap between the upstream ImageNet classification task and the downstream object detection task. Experiments show that FP-DETR not only achieves competitive performance on the challenging COCO dataset, but also gains better robustness against common corruptions and generalization to small-size datasets.

ACKNOWLEDGEMENT

This work is supported by National Key R&D Program of China under Grant 2020AAA0105701, National Natural Science Foundation of China (NSFC) under Grants 61872327, and Major Special Science and Technology Project of Anhui (No. 012223665049). Dr Jing Zhang is supported by ARC FL-170100117.

ETHICS STATEMENT

We acknowledge that all co-authors of this work have read and commit to adhering to the ICLR Code of Ethics.

REPRODUCIBILITY STATEMENT

The authors strive to ensure the reproducibility of the experimental results. Specifically, implementation details are provided in Section 4.1, Section A.1, and Section A.3. The related studies with the same experimental setup are carefully cited and referred to. Moreover, the code will be made publicly available to ensure reproducibility.

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

## A    APPENDIX

### A.1    STRUCTURE OF THE MULTI-SCALE TOKENIZER

Our lightweight multi-scale tokenizer extracts feature maps of 4 different levels to construct the input sequences to the encoder-only transformer. Different from existing detection transformers, which use a sophisticated ResNet-50 backbone for feature extraction, our multi-scale tokenizer simply downsamples the input image to desired feature resolution. Specifically, we perform non-overlapping patch embedding, similar to the seminal ViT (Dosovitskiy et al., 2020). The only difference is that we perform multi-scale tokenization and adopt a hierarchical structure to progressively embed the tokens of large resolution. The detailed structure of our multi-scale tokenizer is shown in Table 6. The dimension $D'$ is 192, 384, and 384, respectively for our Lite, Small, and Base model. After patch embedding, the 2D sequence is flattened to 1D sequence and further projected to tokens with dimension $D$.

Table 6: The architecture of the lightweight multi-scale tokenizer.

| Multi-scale Tokenizer |
| :---: |
| Conv $8 \times 8 \times D'$, stride 8, pad 0 |
| LayerNorm |
| Conv $2 \times 2 \times D'$, stride 2, pad 0 |
| LayerNorm |
| Conv $2 \times 2 \times D'$, stride 2, pad 0 |
| LayerNorm |
| Conv $2 \times 2 \times D'$, stride 2, pad 0 |
| LayerNorm |

### A.2    EXPLORING ENCODER-DECODER STRUCTURE

While most detection transformers contain an encoder-decoder transformer, existing study on vision transformers extensively explores the encoder-only transformer for image classification. This results in a discrepancy between the upstream ImageNet classification task and the downstream object detection task. To mitigate this problem, we first explore pre-training the encoder-decoder transformer on the ImageNet classification task, using existing pre-training technologies (Touvron et al., 2021a). Specifically, we pre-train and fine-tune an encoder-decoder transformer structure, denoted as FP-EncDec, on the ImageNet-1k dataset, follow the same implementation in 4.1. The model follows the design of our FP-DETR-Lite, except the last 6 layers of self-attention layers are replaced by 6 decoder layers and the task adaptor is removed. In each decoder layer, a standard self-attention layer (Vaswani et al., 2017) and a multi-scale deformable attention layer (Zhu et al., 2020) are applied sequentially, following Deformable DETR. Two variants of FP-EncDec are considered: (1) pre-training with a single class token, follow the common practice in training vision transformers (Dosovitskiy et al., 2020); and (2) pre-training with 300 class tokens, which equals the number of object query embeddings for fine-tuning. The 300 class tokens are pooled into a single class token before final classification.

Table 7: Comparison of encoder-decoder transformers and encoder-only transformer for pre-training and fine-tuning.

| Method | Structure | Token Number | Acc@Top-1 | AP | AP$_{50}$ | AP$_{75}$ | AP$_S$ | AP$_M$ | AP$_L$ | Params |
| :--- | :---: | :---: | :---: | :--- | :--- | :--- | :--- | :--- | :--- | :---: |
| FP-EncDec | Enc-Dec | 1 | 72.0 | 33.1 | 51.8 | 35.7 | 17.9 | 35.9 | 43.4 | 12M |
| FP-EncDec | Enc-Dec | 300 | 76.4 | 35.2 | 54.0 | 37.9 | 20.9 | 38.1 | 47.0 | 12M |
| FP-DETR-Lite | Enc | 1 | 77.3 | 37.2 | 56.5 | 40.4 | 21.7 | 40.0 | 48.6 | 11M |

The results for both pre-training on ImageNet-1k and fine-tuning on COCO 2017 dataset are shown in Table 7. As can be seen, pre-training the transformer decoder with a single class token performs poorly on the ImageNet classification task, since both the self-attention layers and the projections on the class token in the cross-attention layer are trained on a single class token, which could easily

lead to overfitting. The low pre-training accuracy also limits the model's detection performance on the downstream task. By contrast, FP-EncDec pre-trained with 300 class tokens performs better on ImageNet-1k classification, the top-1 accuracy is 4.4 higher than the single class token counterpart, since the decoder is more sufficiently trained. However, it is still inferior to our encoder-only FP-DETR-Lite. We conjuncture that this is caused by (1) existing training techniques for vision transformers have been heavily tuned towards the encoder-only structure, which can be sub-optimal for the encoder-decoder transformer; and (2) the discrepancy between upstream and downstream tasks also degenerates the model's performance on object detection, even if the decoder has been pre-trained. We expect these findings may provide useful insights to the community to rethink the current paradigm of pre-training vision transformers, and pay more attention to the pre-training of encoder-decoder transformers.

## A.3 PRE-TRAINING ON IMAGENET-21K

For ImageNet-21k pre-training, we follow the pipeline in Ridnik et al. (2021). The ImageNet-21k is first pre-processed by three steps: (1) cleaning invalid classes; (2) validation split; and (3) image re-sizing. Afterward, the model is trained on the processed dataset (ImageNet-21k-P) using the semantic softmax training (Ridnik et al., 2021). Specifically, our model is pre-trained on ImageNet-21k-P with a batch size of 4096 for 80 epochs. The model is initialized from ImageNet-1K pre-trained weights. The learning rate is initialized as 3e-4 and scheduled using one-cycle policy (Smith, 2018). RandAugment (Cubuk et al., 2020), Cutout (DeVries & Taylor, 2017), Label-smoothing (Szegedy et al., 2016), and True-weight-decay (Loshchilov & Hutter, 2018) are adopted for regularization. For more details on pre-training, please refer to Ridnik et al. (2021).

For fine-tuning FP-DETR further, we adopt the same implementation details in 4.1. For a fair comparison, we also finetuned Deformable DETR (Zhu et al., 2020) with an ImageNet-21k-P pre-trained ResNet-50 backbone. The ResNet-50 pre-trained weight is taken from the official release[2] of Ridnik et al. (2021).

Table 8: Comparision of models fine-tuned from ImageNet-1k and ImageNet-21k pre-trained weights. † indicates the model is pre-trained on ImageNet-21k.

| Method | Structure | Bone | Epochs | AP | $AP_{50}$ | $AP_{75}$ | $AP_S$ | $AP_M$ | $AP_L$ | params |
|---|---|---|---|---|---|---|---|---|---|---|
| Deformable DETR | Enc-Dec | R50 | 50 | 43.8 | 62.6 | 47.7 | 26.4 | 47.1 | 58.0 | 40M |
| Deformable DETR† | Enc-Dec | R50 | 50 | 42.9 | 62.1 | 46.6 | 25.1 | 46.3 | 57.7 | 40M |
| FP-DETR-Base | Enc | - | 50 | 43.3 | 63.9 | 47.7 | 27.5 | 46.1 | 57.0 | 36M |
| FP-DETR-Base† | Enc | - | 50 | 43.7 | 64.1 | 47.8 | 26.5 | 46.7 | 58.2 | 36M |

The result is shown in Table 8. As can be seen, our FP-DETR can benefit from fully pre-training on the larger-scale ImageNet-21k dataset as well as fine-tuning with the task adaptor. Counter-intuitively, the performance of Deformable DETR degenerates when using the weights pre-trained on ImageNet-21k. We conjuncture that only pre-training the CNN backbone of the model makes it difficult for the randomly initialized transformer to adapt to the pre-trained backbone, especially when the backbone has already been well-trained. More research efforts should be made to further dig into this problem.

---

[2]https://github.com/Alibaba-MIIL/ImageNet21K

