# OpenReview forum: "FP-DETR: Detection Transformer Advanced by Fully Pre-training"
_ICLR.cc/2022/Conference — ICLR 2022 Poster_

### Official Review · Reviewer_Jj3q · 2021-10-26

**Correctness:** 3
**Technical Novelty And Significance:** 2
**Empirical Novelty And Significance:** 3
**Recommendation:** 5
**Confidence:** 4

**Main Review:**

## Strengths

+ The submission is clearly written. The motivation and technical details are easy to understand.

+ The authors find the similarity between textual prompt filling task in NLP and object detection and propose a solution to mitigate the gap.

+ The proposed network (FP-DETR) generalizes well on two small datasets (COCO-C and Cityscapes)

## Weakness

- The technical novelty is not fully verified. According to the section 3, the authors first pre-train the encoder part on the ImageNet. In Figure 1, the pre-training leverages class token as a placeholder which could later be replaced by visual prompt information. However, the class token domain has a large gap between the visual domain with visual contents. As a solution, the authors propose a task adapter (which is an self-attention layer or a Bi-LSTM) to mitigate the gap. According to the sentence above eq5, the major function of the prompt encoder is to `improve the modeling of the inter-object relationships`. In such sense, it would be confusing whether the task-adapter is used to better model the object interaction, or to adapt the pre-training image classification task to object detection task. I would suggest the authors to calculate some embedding distance to show that task-adapter-processed embeddings' distances to class token embeddings in pre-training are smaller than pre-task-adaptor embeddings' distances. If the distances are smaller, the task adapter is verified that it could be used to transform the new query embeddings from visual prompts to class token embedding spaces, so that the FP-DETR pre-training could be easily adapted to object detection task.

- The generalization ability is inconsistent with pre-training datasets' sizes. According to the findings in section 4.2, pre-training on ImageNet-21k has lower performance than that from ImageNet-1k. The authors conjectured that the random initialization of transformer may be the cause. However, if the pre-training strategy cannot be scaled to larger datasets to further improve the performance, this method may have smaller scope of application. It may need more efforts to solve the scalability problem.

**Summary Of The Paper:**

This paper proposes a new method of pre-training transformer based object detector (FP-DETR). Compared to previous methods, the authors propose to pre-train DETR's encoder with ConvNet backbones on ImageNet. During the finetuning stage, the authors proposes a task adapter to map the visual prompt to classification token domain to mitigate the gap between image classification and detect object tasks. Experiments on MS-COCO show that the FP-DETR achieves competitive performance and generalizes well on small scale datasets.

**Summary Of The Review:**

This submission provides a novel way of pre-training DETR's encoder part on image classification task first, and then finetuning DETR on object detection task. However, there is one novelty not fully verified. Besides, the method may have some problem in scalability. Thus, I would wait for authors' response before making the decision.

---

> ### Author Response · Authors · 2021-11-19
> **Responses to Reviewer Jj3q (1/2)**
>
> Thanks for your careful review and constructive suggestions. The questions are answered as follows.
>
> ---
>
> **Q#1**: I would suggest the authors to calculate some embedding distance to show that task-adapter-processed embeddings' distances to class token embeddings in pre-training are smaller than pre-task-adaptor embeddings' distances.
>
> **A#1**: Thanks for the good suggestion. To gain a better understanding of the effectiveness of the task adaptor, we perform experiments to measure the embedding distance between the object queries on the downstream task and the class token embedding on the pre-training task.  Since the class token and the object queries at the input level do not yet aggregate context from the input image through the attention mechanism, it's hard to interpret or measure the semantics distances based on these embedding. Thus we choose to measure the semantic distances based on the class token and object queries output by the last transformer layer, where the tokens have already aggregated context from the input image.
>
> Specifically, we adopt **Wasserstein distance** as the distance metric, since it can effectively measure the gap between two discrete distributions. All 10 object classes shared by both ImageNet-1K and COCO 2017 are used to measure the embedding distance. And for each object class, all ground-truth object instances are sampled from each dataset. The class tokens and object query representations for each ground-truth instance are required for the distance calculation. The class tokens can be easily obtained from the output of the last encoder layer. While for the object query, we select the one that best matches the ground-truth object instance from the 300 object queries output by the last encoder layer. L2 distance normalized by feature dimension is adopted as the distance metric between two instances to compute the cost matrix in the optimal transport problem [1]. And all instances are weighted equally.
>
> The Wasserstein distance between the class token embedding in the upstream task and the object queries in the downstream task is shown in the following table, where 'mean' indicates the average Wasserstein distance of all object classes.
>
> | Task Adaptor |   Mean    | parking_meter | zebra | umbrella | kite  |  cup  | banana | orange | broccoli | toaster | vase  |
> | :----------: | :-------: | :-----------: | :---: | :------: | :---: | :---: | :----: | :----: | :------: | :-----: | :---: |
> |     w/o      | **1.338** |     1.324     | 1.185 |  1.413   | 1.356 | 1.453 | 1.268  | 1.284  |  1.223   |  1.397  | 1.472 |
> |      w       | **1.135** |     1.247     | 1.089 |  1.338   | 1.301 | 1.397 | 1.201  | 1.194  |  1.095   |  1.314  | 1.417 |
>
> As can be seen, the distribution gap between the embedding in the upstream task and the embedding in the downstream task is smaller when the task adaptor is applied. The remained distribution gap can stem from (1) the inherent difference between the object instances from the ImageNet-1K and COCO 2017 dataset; (2) the feature representation is slightly drifted during the fine-tuning processing.
>
> [1] Optimal transport: old and new, Springer 2009.

---

> > ### Author Response · Authors · 2021-11-19
> > **Responses to Reviewer Jj3q (2/2)**
> >
> >
> >
> > ---
> >
> > **Q#2**: The generalization ability is inconsistent with pre-training datasets' sizes. According to the findings in section 4.2, pre-training on ImageNet-21K-P has lower performance than that from ImageNet-1K.
> >
> > **A#2**: Actually, our **FP-DETR** enjoys a larger scale pre-training. As shown in Table 2, the pre-training on ImageNet-21K-P makes our FP-DETR-Base† 0.4 AP higher than its ImageNet-1K pre-trained counterpart.
> >
> > In section 4.2 and Appendix A.3, we found that the performance of **Deformable DETR**, the state-of-the-art detection transformer, is slightly worse when changing the ImageNet-1K pre-trained backbone to the ImageNet-21K-P pre-trained backbone. We conjuncture that only pre-training the CNN backbone of the model makes it difficult for the randomly initialized transformer to adapt to the well pre-trained backbone. More research efforts should be made to further dig into this problem. Since this is not the focus of this paper, we will leave this problem for future study.
> >
> > To further alleviate the reviewer's concern, we also added the generalization and robustness analysis of **FP-DETR-Base†** pre-trained on ImageNet-21K-P. The results are shown in the following tables. As can be seen, both the robustness and generalization ability of our FP-DETR can be further improved by pre-training on a larger scale dataset. Specifically, 0.7 and 0.9 performance gains are achieved on Cityscapes and COCO-C datasets, respectively.
> >
> > |    Method     | Structure | Epochs |  AP  | AP50 | AP75 | AP@S | AP@M | AP@L | Params |
> > | :----------- | :-------: | :----: | :--: | :--: | :--: | :--: | :--: | :--: | :----: |
> > | FP-DETR-Base  |    Enc    |   50   | 31.8 | 56.4 | 31.2 | 11.6 | 33.2 | 51.1 |  36M   |
> > | FP-DETR-Base† |    Enc    |   50   | 32.5 | 58.9 | 31.5 | 13.3 | 35.1 | 53.0 |  36M   |
> >
> >
> > |    Method     |   Mean   |   Gauss   |  Shot   |   Impul    |   Defocus    |    Glass    |  Motion   |   Zoom   |
> > | :----------- | :------: | :-------: | :-----: | :--------: | :----------: | :---------: | :-------: | :------: |
> > | FP-DETR-Base  |   23.7   |   22.5    |  22.7   |    20.4    |     22.9     |    18.9     |   21.7    |   8.0    |
> > | FP-DETR-Base† |   24.6   |   22.7    |  22.8   |    20.8    |     23.3     |    19.6     |   22.7    |   8.6    |
> > |  **Method**   | **Snow** | **Frost** | **Fog** | **Bright** | **Contrast** | **Elastic** | **Pixel** | **JPEG** |
> > | FP-DETR-Base  |   22.4   |   27.1    |  33.4   |    38.0    |     25.3     |    26.1     |   20.0    |   25.6   |
> > | FP-DETR-Base† |   24.1   |   28.2    |  34.7   |    38.7    |     27.1     |    28.3     |   21.4    |   25.6   |
> >
> > To summarize, our **FP-DETR** is scalable to pre-training at a large scale. We will improve the writing in section 4.2 and Appendix A.3 to make it more clarified.

---

### Official Review · Reviewer_s2Cu · 2021-11-03

**Correctness:** 3
**Technical Novelty And Significance:** 2
**Empirical Novelty And Significance:** 3
**Recommendation:** 6
**Confidence:** 4

**Main Review:**

Overall, I like the paper but I feel hesitant to accept this paper at its current status . While it is no secret that one can improve model robustness and generalizability through pre-training, I do believe that the method contributes values to the community with an architecture that is designed to better utilize imageNet classification pre-training and close the gap between classification and detection in the fine-tuning stage. However, I found the current experimental protocol is not solid enough to support some of the claimed contributions. I am happy to change my rating if the author could provide justification through discussion or additional experiments.

Pros:

- The paper is well written and easy to read. The intuition behind the algorithm design is sufficiently discussed. The technical session is described in detail.
- I like the idea of decreasing the model discrepancy between upstream classification tasks and downstream detection tasks through an encoder only transformer.
- The intuition to provide location attention and inter-object relationship attention is interesting
- The overall validity of the algorithm is backed by performance close to SOTA on COCO and outstanding in Cityscapes.

Cons:
- The main concern I have in the experiment is whether we can decouple the advantage of the proposed architecture or the fact that the Transformer encoder is being pretrained when compared to other SOTA (Table 2). E.g. What’s the performance of the proposed algorithm with normal ResNet backbones? What’s the performance of Deformable Tranformer with Encoder pretrained？
- The ablation analysis on the lite version (Table 3) is not as strong as one in the base version. It is easier to improve on top of a weak performance.


**Summary Of The Paper:**

This paper proposes a novel transformer-based detection algorithm (FP-DETR) that could benefit from large-scale pre-training for model robustness and generalization on small datasets. FP-DETR consists of a encoder only transformer on top of a set of lightweight feature extracting layers for pre-training. It is then fine-tuned with a task-adapter that helps the model attending to object location as well as be aware of inter-object relationship for detection task. Evaluation is done on both large scale dataset COCO and relatively small domain CityScapes with ablation analysis on different system modules and design choices. The result indicates that FP-DETR presents better generalization capabilities on smaller dataset with on-part performance compared to other SOTA in general settings.


**Summary Of The Review:**

The paper proposed a novel architecture to empower a transformer-based object detector with smoother pre-training and fine-tuning paradigm. The paper is well-written and provides interesting discussion on its intuition and new perspective on positional embeddings in this context. However, pre-training helps improve the model is not a novel theory by itself. The insufficient experimental comparison decreases the strength of the paper. I am looking forward to seeing the rebuttal and willing to change my rating accordingly.

---

> ### Author Response · Authors · 2021-11-19
> **Responses to Reviewer s2Cu**
>
> Thanks for your careful review and constructive suggestions. The questions are answered as follows.
>
> ---
>
> **Q#1: whether we can decouple the advantage of the proposed architecture or the fact that the Transformer encoder is being pretrained when compared to other SOTA (Table 2). E.g. What’s the performance of the proposed algorithm with normal ResNet backbones? What’s the performance of Deformable Transformer with Encoder pretrained？**
>
> **A#1**: Thanks for your good suggestion. We added ablated experiments to decouple the effectiveness of the proposed architecture and our fully pre-training paradigm, as shown in the following table.
>
> Firstly, to ablate the effectiveness of our proposed architecture, we train the lite version of our model with a ResNet-50 backbone. To eliminate the influence of our full-network pre-training paradigm, we follow Deformable DETR to use the ImageNet-1k pre-trained ResNet-50 while leaving the transformer encoder trained from scratch. We denote this model as FP-DETR-R50 (P) where P indicates the model is partially pre-trained. From the first two lines in the following table, it can be seen that the transformer encoder structure obtains an on-par performance compared with the transformer encoder-decoder structure. However, it should be noted that the encoder-only structure allows us to fully pre-train the detection transformer on the common ImageNet classification task.
>
> Secondly, to investigate the effectiveness of our fully pre-training paradigm, we pre-train and fine-tune the lite version of our model with a ResNet-50 backbone. The model is denoted as FP-DETR-R50. Following the reviewer's suggestion, we also fine-tune a Deformable DETR variant with both ResNet-50 backbone and transformer encoder jointly pre-trained. It can be seen that Deformable-DETR with encoder pre-trained is slightly better than Deformable-DETR, thanks to the pre-training of the transformer encoder. However, the improvement is less significant than FP-DETR-R50. We conjuncture that the left 6-layer randomly initialized decoder still holds back the model convergence and limits the final performance. By contrast, FP-DETR-R50 can take advantage of our full-network pre-training, and achieves 0.6 AP gain compared with Deformable DETR.
>
> |     Method      | Pre-trained Part | Structure | Backbone  |  AP  | AP50 | AP75 | AP@S | AP@M | AP@L | Params |
> | :------------- | :--------------: | :-------: | :-------: | :--: | :--: | :--: | :--: | :--: | :--: | :----: |
> | Deformable-DETR |     backbone     |  Enc-Dec  | ResNet-50 | 43.8 | 62.6 | 47.7 | 26.4 | 47.1 | 58.0 |  40M   |
> | FP-DETR-R50 (P) |     backbone     |    Enc    | ResNet-50 | 43.8 | 63.0 | 47.6 | 26.2 | 47.3 | 58.2 |  40M   |
> | Deformable-DETR |  backbone + Enc  |  Enc-Dec  | ResNet-50 | 44.0 | 63.2 | 47.7 | 26.7 | 47.0 | 58.7 |  40M   |
> |   FP-DETR-R50   |  backbone + Enc  |    Enc    | ResNet-50 | 44.4 | 63.5 | 48.5 | 26.9 | 47.6 | 59.6 |  40M   |
>
> To summarize, though the encoder-only structure does not directly improve the model performance, it allows us to take advantage of the fully-network pre-training, and thereby achieve state-of-the-art object detection performance.
>
> ---
>
> **Q#2**: The ablation analysis on the lite version (Table 3) is not as strong as one in the base version.
>
> **A#2**: Thanks for your good suggestion. To better understand the effectiveness of our task adaptor, we also perform ablation analysis on the base variant of our FP-DETR. The result is shown in the following table. It is true that the performance gap on the lite version is more significant than the base version, e.g., the performance gain brought by the default task adaptor implemented as the shared self-attention is 4.4 v.s. 3.7 for the lite and base variants, respectively.
>
> But still, the result on the base model is consistent with our ablation study based on the lite version. Specifically, both bidirectional LSTM and self-attention layer can be used to improve the relationship modeling between the visual prompts, though the self-attention layer performs better. Moreover, the task adaptor works slightly better when it is shared across layers.
>
> |  Task Adaptor  | Shared |  AP  | AP50 | AP75 | AP@S | AP@M | AP@L | Params |
> | :------------: | :----: | :--: | :--: | :--: | :--: | :--: | :--: | :----: |
> |      w/o       |  N/A   | 39.6 | 58.7 | 43.2 | 24.6 | 42.9 | 51.2 |  35M   |
> |    Bi-LSTM     |   no   | 41.1 | 59.9 | 44.9 | 24.6 | 44.5 | 53.5 |  46M   |
> |    Bi-LSTM     |  yes   | 41.3 | 60.4 | 45.0 | 25.5 | 44.4 | 53.6 |  37M   |
> | Self-attention |   no   | 42.8 | 63.2 | 46.7 | 26.1 | 46.0 | 56.6 |  40M   |
> | Self-attention |  yes   | 43.3 | 63.9 | 47.7 | 27.5 | 46.1 | 57.0 |  36M   |

---

### Official Review · Reviewer_JhDk · 2021-11-04

**Correctness:** 4
**Technical Novelty And Significance:** 2
**Empirical Novelty And Significance:** 3
**Recommendation:** 6
**Confidence:** 4

**Main Review:**

- This work introduced the ‘prompting’ to detection task. By mitigating the gap between pre-training and finetuning task, it shows a better performance on COCO dataset. It is also better on robustness and gereralization to small-size dataset. Authors conducted ablation studies to verify the different settings.

- In Table 7, it shows the FP-EncDec with 300 tokens performs better than that with only 1 token on pre-training and fine-tuning tasks. I wonder if we also apply 300 tokens to FP-DETR on the pretraining, will performance of pretraining and downstream tasks become better as well? It is interesting to see it as it may provide more flexibility to different downstream tasks.
- In table 2, is it possible to have an ablation study in which we have FP-DETR with ResNet-50 backbone?

- Minor: it seems there is a typo -  PT-DETR in the introduction?


**Summary Of The Paper:**

This work proposes a new transformer framework for detection task by masking better use of features in pretraining phase. Inspired by ‘prompting’ in the NLP task, it treats query positional embeddings as visual prompts to better localize objects. With the similar architecture design, it mitigates the discrepancy between pretraining and finetuning stages. Task adaptor is introduced to capture inter-object relationships. The proposed framework achieves competitive performance and becomes more robust and generalizable.

**Summary Of The Review:**

Although the idea is borrowed from the NLP field, applying it to the CV field may be helpful for the community. It provides a new perspective towards the mitigate the discrepancy between pre-training and finetuning. At the current stage, I prefer to accept this work.

---

> ### Author Response · Authors · 2021-11-19
> **Responses to Reviewer JhDk**
>
> Thanks for your careful review and constructive suggestions. The questions are answered as follows.
>
> ---
>
> **Q#1**: I wonder if we also apply 300 tokens to FP-DETR on the pretraining, will performance of pretraining and downstream tasks become better as well?
>
> **A#1**: Thanks for the good suggestion. We followed your advice to pre-training and fine-tuning our FP-DETR-Lite and FP-DETR-Base with 300 tokens. The result is shown in the 4th and 6th row in the following table (shown in bold). It's very interesting to see that the performances of both ImageNet classification and downstream object detection are slightly improved.
>
> We suppose the performance gain on ImageNet classification comes from the aggregation of more context for final prediction. Specifically, with the deformable attention, each class token only aggregates sparse context from the whole image. As the number of class tokens rises from 1 to 300, the final prediction can be made based on a more comprehensive context sampling from the input image.
>
> The performance gain on the downstream object detection task can be attributed to two aspects. Firstly and obviously, better pre-training help the model learn better feature representation for the downstream task [1]. Secondly, since the number of tokens for both pre-training and downstream tasks is kept the same, the model does not have to learn the query embedding from scratch on the downstream task.
>
> It's noteworthy that pre-training FP-DETR with 300 tokens brings less significant performance gain compared with the FP-EncDec counterpart. This is because the transformer decoder in FP-EncDec can hardly be sufficiently pre-trained with a single class token, and improving the class token number help the pre-training of the decoder. By contrast, our FP-DETR adopts the transformer encoder architecture, which can easily be pre-trained even with only a single class token.
>
> |      Method      | Structure | #Token  | Acc@Top-1 |    AP    |   AP50   |   AP75   |   AP@S   |   AP@M   |   AP@L   | Params  |
> | :-------------- | :-------: | :-----: | :-------: | :------: | :------: | :------: | :------: | :------: | :------: | :-----: |
> |    FP-EncDec     |  Enc-Dec  |    1    |   72.0    |   33.1   |   51.8   |   35.7   |   17.9   |   35.9   |   43.4   |   12M   |
> |    FP-EncDec     |  Enc-Dec  |   300   |   76.4    |   35.2   |   54.0   |   37.9   |   20.9   |   38.1   |   47.0   |   12M   |
> |   FP-DETR-Lite   |    Enc    |    1    |   77.3    |   37.2   |   56.5   |   40.4   |   21.7   |   40.0   |   48.6   |   11M   |
> | **FP-DETR-Lite** |  **Enc**  | **300** | **77.6**  | **37.4** | **57.0** | **40.6** | **20.5** | **40.6** | **51.4** | **11M** |
> |   FP-DETR-Base   |    Enc    |    1    |   82.0    |   43.3   |   63.9   |   47.7   |   27.5   |   46.1   |   57.0   |   36M   |
> | **FP-DETR-Base** |  **Enc**  | **300** | **82.2**  | **43.4** | **63.6** | **47.6** | **26.2** | **46.4** | **57.4** | **36M** |
>
> [1] Do Better ImageNet Models Transfer Better? CVPR 2019.
>
> ---
>
> **Q#2: In table 2, is it possible to have an ablation study in which we have FP-DETR with ResNet-50 backbone?**
>
> **A#2**: Thanks for your good suggestion. We trained the lite version of our FP-DETR with a ResNet-50 backbone, since its number of parameters is similar to Deformable DETR with ResNet-50 backbone. The newly added result is shown in bold in the following table.
>
> |     Method      |  Pre-trained Part  | Structure |   Backbone    |    AP    |   AP50   |   AP75   |   AP@S   |   AP@M   |   AP@L   | Params  |
> | :------------- | :----------------: | :-------: | :-----------: | :------: | :------: | :------: | :------: | :------: | :------: | :-----: |
> | Deformable-DETR |      backbone      |  Enc-Dec  |   ResNet-50   |   43.8   |   62.6   |   47.7   |   26.4   |   47.1   |   58.0   |   40M   |
> | **FP-DETR-R50** | **backbone + Enc** |  **Enc**  | **ResNet-50** | **44.4** | **63.5** | **48.5** | **26.9** | **47.6** | **59.6** | **40M** |
>
> As can be seen, FP-DETR-Lite with ResNet-50 backbone is 0.6 AP better than the state-of-the-art Deformable DETR on COCO 2017 validation set. This ablated experiment further validates the benefit of our fully pre-training paradigm. We will add the result to the final version of our paper.
>
> ---
>
> **Q#3**: Minor: it seems there is a typo - PT-DETR in the introduction?
>
> **A#3**: We are sorry for the mistake. We will fix the typo and double-check our manuscript.

---

### Official Review · Reviewer_MjUh · 2021-11-09

**Correctness:** 3
**Technical Novelty And Significance:** 2
**Empirical Novelty And Significance:** 2
**Recommendation:** 6
**Confidence:** 5

**Main Review:**

This paper is well-organized and easy to follow.

Strength:
+ Encoder-only transformer design without the need of extra visual backbone
+ Lower parameters compared to concurrent works

Weakness:
- The effeteness of pre-training is not well justified. The paper claims "large-scale pre-training on ImageNet helps FP-DETR learn more generalizable representation." However, the author only conducts experiments on imagenet-1k, which barely counts a large-scale pre-training. In order to further prove the benefits, it would be best to re-run experiments on imagenet-22k and show the continuous improvement .
- The necessary of deformable convolution layer.  The encoder-only transformer design would be more appreciated if not using deformable convolution. How big will the impact be if trained without deformable convolution?
- Latency. Since author focuses on small models, it would be best to report latency.
- The robustness analysis is not convincing and not related to the main contribution. The author should include more ablation studies on hyper-parameters instead (such as #encoder, #dim, #token, etc)

**Summary Of The Paper:**

This paper argues that recently developed detection transformers didn't employ pre-training on transformer layers and not benefit from pretraining. Hence, it proposes FP-DETR to fully pretrain transformer on image classification and then finetune it for object detection by changing the task adapter. Compared to previous methods, it only uses encoder-only transformers. Further experiments show reasonable performances and robustness analysis.

**Summary Of The Review:**

This paper shares some insights on the proposed FP-DETR, but it lacks comprehensive experiments to justify the claims. After reading the rebuttal and extra ablation studies, most of my questions have been answered. I believe this paper has some merit and worth accepting for a poster. Hence, I have raised my score.

---

> ### Author Response · Authors · 2021-11-19
> **Responses to Reviewer MjUh (1/4)**
>
> Thanks for your careful review and constructive suggestions. The questions are answered as follows.
>
> ---
>
> **Q#1**: In order to further prove the benefits, it would be best to re-run experiments on imagenet-22K and show the continuous improvement.
>
> **A#1**: Thanks for your good suggestion. Since most object detection methods are pre-trained on ImageNet-1k, we mainly follow this paradigm for a fair comparison. As the reviewer suggested, it would be better to validate the effectiveness of the fully pre-training on a larger scale dataset. Thus, we conducted experiments on the ImageNet-21K-P [1] dataset for a better understanding of our method.
>
> The ImageNet-21K-P is a cleaned version of the ImageNet-22K (ImageNet-21K) dataset, it contains a similar amount of images as the ImageNet-22K dataset (12,358,688 v.s. 14,197,122), roughly ten times the number of images in ImageNet-1K. Moreover, it has been proved effective for transferring to downstream tasks, and thus has been included in the official ImageNet website at https://image-net.org/download-images.php. Please refer to our appendix and their paper for more details.
>
> As can be seen from Table 2 and Table 8 in Appendix A.3, our fully pre-training can benefit from the pre-training at a larger scale (+0.4 AP). To further verify the effectiveness of larger-scale pre-training, we also provide the generalization and robustness analysis of our FP-DETR-Base† pre-trained on ImageNet-21K-P. Experiment results on Cityscapes and COCO-C are shown in the following two tables, respectively.
>
> |    Method     | Structure | Epochs |    AP    | AP50 | AP75 | AP@S | AP@M | AP@L | Params |
> | :----------- | :-------: | :----: | :------: | :--: | :--: | :--: | :--: | :--: | :----: |
> | FP-DETR-Base  |    Enc    |   50   | **31.8** | 56.4 | 31.2 | 11.6 | 33.2 | 51.1 |  36M   |
> | FP-DETR-Base† |    Enc    |   50   | **32.5** | 58.9 | 31.5 | 13.3 | 35.1 | 53.0 |  36M   |
>
>
> |    Method     |   Mean   |   Gauss   |  Shot   |   Impul    |   Defocus    |    Glass    |  Motion   |   Zoom   |
> | :----------- | :------: | :-------: | :-----: | :--------: | :----------: | :---------: | :-------: | :------: |
> | FP-DETR-Base  | **23.7** |   22.5    |  22.7   |    20.4    |     22.9     |    18.9     |   21.7    |   8.0    |
> | FP-DETR-Base† | **24.6** |   22.7    |  22.8   |    20.8    |     23.3     |    19.6     |   22.7    |   8.6    |
> |  **Method**   | **Snow** | **Frost** | **Fog** | **Bright** | **Contrast** | **Elastic** | **Pixel** | **JPEG** |
> | FP-DETR-Base  |   22.4   |   27.1    |  33.4   |    38.0    |     25.3     |    26.1     |   20.0    |   25.6   |
> | FP-DETR-Base† |   24.1   |   28.2    |  34.7   |    38.7    |     27.1     |    28.3     |   21.4    |   25.6   |
>
> As can be seen, FP-DETR-Base† shows better generalization and robustness ability compared to the ImageNet-1K pre-trained counterpart, achieving 0.7 AP and 0.9 AP gain on Cityscapes and COCO-C datasets, respectively. These results demonstrate the benefits of our full-network pre-training paradigm, and that FP-DETR can take advantage of pre-training at a larger scale.
>
> [1] ImageNet-21K Pretraining for the Masses, NeurIPS 2021.
>
> ------
>
> **Q#2: The necessary of deformable convolution layer.**
>
> **A#2**: The input images for object detection are often in much higher resolution (e.g. 800*1333) compared with the image size for classification (e.g. 224 *224). Under this condition, utilizing the vanilla attention mechanism is very expensive due to its quadratic complexity. As a result, we can hardly scale the model to the small or base version. Thus we prefer to implement our FP-DETR with deformable attention.
>
> -----

---

> > ### Author Response · Authors · 2021-11-19
> > **Responses to Reviewer MjUh (2/4)**
> >
> > **Q#3: Since author focuses on small models, it would be best to report latency.**
> >
> > **A#3**: Thanks for your good suggestion, we follow the convention in DETR to measure the inference latency of FP-DETR based on the first 100 images of COCO val split. All models are tested on the NVIDIA A100 GPU and the results are shown in the following table.
> >
> > |     Method      | Epochs |  AP  | AP50 | AP75 | AP@S | AP@M | AP@L | Params | Latency | FPS  |
> > | :---- | :----: | :--: | :--: | :--: | :--: | :--: | :--: | :----: | :-----: | :--: |
> > |      DETR       |  500   | 42.0 | 62.4 | 44.2 | 20.5 | 45.8 | 61.1 |  41M   |  26ms   |  38  |
> > |    DETR-DC5     |  500   | 43.3 | 63.1 | 45.9 | 22.5 | 47.3 | 61.1 |  41M   |  45ms   |  22  |
> > | Deformable-DETR |   50   | 43.8 | 62.6 | 47.7 | 26.4 | 47.1 | 58.0 |  40M   |  39ms   |  26  |
> > |  FP-DETR-Lite   |   50   | 37.2 | 56.5 | 40.4 | 21.7 | 40.0 | 48.6 |  11M   |  32ms   |  31  |
> > |  FP-DETR-Small  |   50   | 42.5 | 62.6 | 45.9 | 25.3 | 45.5 | 56.9 |  24M   |  41ms   |  24  |
> > |  FP-DETR-Base   |   50   | 43.3 | 63.9 | 47.7 | 27.5 | 46.1 | 57.0 |  36M   |  50ms   |  20  |
> > |  FP-DETR-Base†  |   50   | 43.7 | 64.1 | 47.8 | 26.5 | 46.7 | 58.2 |  36M   |  50ms   |  20  |
> >
> > As can be seen, FP-DETR-Lite runs faster and FP-DETR-Small is on par with Deformable DETR in latency. FP-DETR-Base runs at 50 ms per image, and is 11 ms slower than Deformable DETR. In other words, it's about 20% slower than Deformable DETR. The latency can be further reduced by aggregating multi-scale context into feature at a single scale or improving the hardware unit support to accelerate the deformable attention mechanism, which we will continue to optimize in our future work.
> >
> > ----
> >
> > **Q#4**: The robustness analysis is not convincing and not related to the main contribution.
> >
> > **A#4**: For the robustness analysis, Hendrycks et. al. [2] propose a benchmark for neural network robustness to common corruptions, and the benchmark is widely accepted in the research area to facilitate comparison of model robustness. We strictly follow the pipeline in [2, 3] and generate 15 types of corrupted images and 5 levels of severity each, to measure the performance of different models under corruption. The same benchmark has also been adopted in many other works to perform robustness analysis, including NoisyStudent [4], SupCon [5], SegFormer [6], etc. Thus we believe the robustness analysis can provide insights into the robustness of different object detection methods against common corruptions.
> >
> > The main contribution of this paper is to improve the detection transformer with fully pre-training. And an immediate benefit of the pre-training is to improve the model robustness [7]. As can be seen from Table 4, FP-DETR-Base is much more robust than existing detection transformers under common corruptions, though it's only on par with the Deformable DETR on the curated COCO 2017 validation set.
> >
> > To summarize, the robustness analysis helps demonstrate the benefit of introducing fully pre-training and is thus closely related to our main contribution.
> >
> > [2] Benchmarking Neural Network Robustness to Common Corruptions and Perturbations, ICLR 2019.
> >
> > [3] Benchmarking Robustness in Object Detection: Autonomous Driving when Winter is Coming, NeurIPS 2019 workshop.
> >
> > [4] Self-training with Noisy Student improves ImageNet classification, CVPR 2020.
> >
> > [5] Supervised Contrastive Learning, NeurIPS 2020.
> >
> > [6] SegFormer: Simple and Efficient Design for Semantic Segmentation with Transformers, NeurIPS 2021.
> >
> > [7] Using Pre-Training Can Improve Model Robustness and Uncertainty, ICML 2019.

---

> > > ### Author Response · Authors · 2021-11-19
> > > **Responses to Reviewer MjUh (3/4)**
> > >
> > > **Q#5**: The author should include more ablation studies on hyper-parameters instead (such as #encoder, #dim, #token, etc)
> > >
> > > **A#5**: Thanks for your good suggestion. The hyper-parameters in FP-DETR mainly follow the design of the existing vision transformer [8, 9]. As the reviewer suggests, we added ablation studies on the number of encoder layers, the number of class tokens for pre-training, and the number of object queries (visual prompts) for fine-tuning. We will discuss them in detail respectively.
> > >
> > > **Firstly**, we added ablated experiments to see the effectiveness of varying the number of encoder layers. The experiment is based on the lite version of FP-DETR, and the results are shown in the following table. It can be seen that reducing the number of encoder layers from 12 to 8 degenerates the model performance, since the model capacity is reduced. By contrast, the performance gain is less significant when increasing the number of encoder layers from 12 to 16. We conjuncture that the deep transformer model may face optimization difficulty and a more sophisticated training paradigm [10, 11] may be required to mitigate this problem. We will add these results to our paper and scale FP-DETR to a larger depth.
> > >
> > > | #Layer | Acc@Top-1 |    AP    | AP50 | AP75 | AP@S | AP@M | AP@L | Params | Latency |
> > > | :----: | :-------: | :------: | :--: | :--: | :--: | :--: | :--: | :----: | :-----: |
> > > | **8**  |   73.6    | **32.9** | 52.4 | 34.7 | 17.5 | 35.8 | 45.1 |   8M   |  24ms   |
> > > | **12** |   77.3    | **37.2** | 56.5 | 40.4 | 21.7 | 40.0 | 48.6 |  11M   |  32ms   |
> > > | **16** |   78.6    | **38.3** | 58.6 | 42.0 | 21.9 | 42.4 | 51.6 |  14M   |  41ms   |
> > >
> > > **Secondly**, we add an ablation study on the number of object query tokens during fine-tuning, as shown in the following table. Both models share the same pre-trained weights. It can be seen that reducing the number of object queries from 300 to 100 degenerates the model's performance by 2.1 AP. The insufficient amount of object queries make the model miss more foreground objects. The result is consistent with the findings in Deformable DETR [12] and Sparse RCNN [13].
> > >
> > > |    Method    | Structure | #Token (fine-tune) |    AP    | AP50 | AP75 | AP@S | AP@M | AP@L | Params |
> > > | :----------: | :-------: | :----------------: | :------: | :--: | :--: | :--: | :--: | :--: | :----: |
> > > | FP-DETR-Lite |    Enc    |      **300**       | **37.2** | 56.5 | 40.4 | 21.7 | 40.0 | 48.6 |  11M   |
> > > | FP-DETR-Lite |    Enc    |      **100**       | **35.1** | 52.6 | 37.5 | 20.3 | 37.3 | 46.6 |  11M   |

---

> > > > ### Author Response · Authors · 2021-11-19
> > > > **Responses to Reviewer MjUh (4/4)**
> > > >
> > > > **Thirdly**, we also perform an ablation study on the number of class tokens during pre-training. The number of object query tokens for fine-tuning is kept as 300. The results are shown in the following table, and the newly added results are shown in bold. As can be seen, both ImageNet classification and object detection performances are marginally improved when pre-training with a larger number of object queries.
> > > >
> > > > We suppose the performance gain on ImageNet classification comes from the aggregation of more context for final prediction. Specifically, with the deformable attention, each class token only aggregates sparse context from the whole image. As the number of class tokens rises from 1 to 300, the final prediction can be made based on a more comprehensive context sampling from the input image.
> > > >
> > > > The performance gain on the downstream object detection task can be attributed to two aspects. Firstly and obviously, better pre-training help the model learn better feature representation for the downstream task [14]. Secondly, since the number of tokens for both pre-training and downstream tasks is kept the same, the model does not have to learn the query embedding from scratch on the downstream task.
> > > >
> > > > |      Method      | Structure | #Token  | Acc@Top-1 |    AP    |   AP50   |   AP75   |   AP@S   |   AP@M   |   AP@L   | Params  |
> > > > | :-------------- | :-------: | :-----: | :-------: | :------: | :------: | :------: | :------: | :------: | :------: | :-----: |
> > > > |    FP-EncDec     |  Enc-Dec  |    1    |   72.0    |   33.1   |   51.8   |   35.7   |   17.9   |   35.9   |   43.4   |   12M   |
> > > > |    FP-EncDec     |  Enc-Dec  |   300   |   76.4    |   35.2   |   54.0   |   37.9   |   20.9   |   38.1   |   47.0   |   12M   |
> > > > |   FP-DETR-Lite   |    Enc    |    1    |   77.3    |   37.2   |   56.5   |   40.4   |   21.7   |   40.0   |   48.6   |   11M   |
> > > > | **FP-DETR-Lite** |  **Enc**  | **300** | **77.6**  | **37.4** | **57.0** | **40.6** | **20.5** | **40.6** | **51.4** | **11M** |
> > > > |   FP-DETR-Base   |    Enc    |    1    |   82.0    |   43.3   |   63.9   |   47.7   |   27.5   |   46.1   |   57.0   |   36M   |
> > > > | **FP-DETR-Base** |  **Enc**  | **300** | **82.2**  | **43.4** | **63.6** | **47.6** | **26.2** | **46.4** | **57.4** | **36M** |
> > > >
> > > > It's noteworthy that pre-training FP-DETR with 300 tokens brings less significant performance gain compared with the FP-EncDec counterpart. This is because the transformer decoder in FP-EncDec can hardly be sufficiently pre-trained with a single class token, and improving the class token number help the pre-training of the decoder. By contrast, our FP-DETR adopts the transformer encoder architecture, which can easily be pre-trained even with only a single class token.
> > > >
> > > > [8] An Image is Worth 16x16 Words: Transformers for Image Recognition at Scale, ICLR 2021.
> > > >
> > > > [9] Training data-efficient image transformers & distillation through attention, ICML 2021.
> > > >
> > > > [10] CaiT: Going deeper with Image Transformers, ICCV 2021.
> > > >
> > > > [11] How to train your ViT? Data, Augmentation, and Regularization in Vision Transformers, Arxiv 2021.
> > > >
> > > > [12] Deformable DETR: Deformable Transformers for End-to-End Object Detection, ICLR 2021.
> > > >
> > > > [13] Sparse R-CNN: End-to-End Object Detection with Learnable Proposals, CVPR 2020.
> > > >
> > > > [14] Do Better ImageNet Models Transfer Better? CVPR 2019.

---

### Decision · Program_Chairs · 2022-01-20

**Decision:**

Accept (Poster)

**Comment:**

Summary: Authors present an approach for transformer based object detection that “fully pretrains” the encoder structure of the transformer, and drops the pretrained convolutional backbone used in other works.

Pros:
- Eliminates need of extra visual backbone
- Fewer parameters than other works
- Achieves competitive performance, especially controlling for model size

Cons:
- Multiple reviewers raised concerns about authors only evaluating their approach with pretraining from ImageNet 1K, which is not considered large scale. Authors replied with new experimental data including pretraining from ImageNet 22k, which improved results.
- Multiple reviewers raised concerns about the need for more ablation experiments, which the authors addressed.

Reviewer scores lean toward accept. Those that lean toward reject raised issues that the authors appear to have addressed sufficiently. Not all reviewers have replied to authors, though understood at least some reviewers on this paper are on end-of-year time-off.

Overall recommendation: accept.